# The Impact of Clinical Pilates Exercises on Tension-Type Headaches: A Case Series

**DOI:** 10.3390/bs13020105

**Published:** 2023-01-27

**Authors:** Agathe Leite, Antoine Matignon, Léa Marlot, Ana Coelho, Sofia Lopes, Gabriela Brochado

**Affiliations:** 1Departamento de Tecnologias de Diagnóstico e Terapêutica, Escola Superior de Tecnologias da Saúde do Tâmega e Sousa, Instituto Politécnico de Saúde do Norte (IPSN), CESPU, 4585-116 Gandra, Portugal; 2Departamento de Fisioterapia, ESS Porto Escola Superior de Saúde, Politécnico do Porto, 4200-072 Porto, Portugal

**Keywords:** disability, functionality, pain, Pilates, quality of life

## Abstract

Tension-type-headaches appear as the third most common disease in the general population and have a big impact on quality of life. The aim is to verify the impact of clinical Pilates exercises on pain intensity, impact headaches, neck disability, state of anxiety, depression, stress and quality of sleep in higher-education students. After a sample selection questionnaire was completed, a series of cases was carried out with 9 undergraduate students with tension-type headaches. Five instruments were applied before and after an intervention program: (i) Numerical Rating Scale—NRS, (ii) Headache Impact Test—HIT-6™, (iii) Neck Disability Index, (iv) Depression, Anxiety, Stress Scales and (v) Pittsburgh Sleep Quality Index. Four sessions of 30 min classes of Pilates exercises were held for 2 weeks. The pain intensity decreased in only 2 participants. The impact of tension-type headaches on normal daily life and ability to function was improved in all patients, and only one patient’s functional disability caused by pain in the cervical region did not improve. In regard to negative emotional states, 6 individuals reported improvements, and only one individual reported a lower quality of sleep after the program. The exercise program can induce positive effects on disorders associated with tension-type-headaches.

## 1. Introduction

Headache is defined as a discomfort or painful sensation located at the level of the head above the orbitomeatal line [1,2]. This disorder constitutes a public health problem due to its high prevalence and high impact on quality of life. In fact, tension-type headache (TTH) and migraine, according to the International Classification of Headache Disorders [1], are respectively the third and sixth most common diseases in the world [1,2,3]. The ICHD-3 is the classification most used by health professionals and describes three types of headache disorders: (i) primary headaches, including migraine, TTH, and trigeminal autonomic headaches; (ii) secondary headaches associated with injury, trauma, vascular and nonvascular infections; and (iii) painful cranial neuropathies and other facial pain. Primary headaches are disorders that exist without an apparent underlying cause. Diagnosis is based on the individual’s description of pain and on meeting the diagnostic criteria set out in the ICHD-3. In contrast, secondary headaches are caused by, among other factors, whiplash, craniotomy, or sleep apnea [1,4]. TTH is described as the most common in adults, affecting 30 to 78% of the population over a lifetime, with this percentage varying worldwide [1]. In Europe, the prevalence is very high, affecting about 60% of the population [5], and this is higher in females (4:5 ratio) [5]. In addition to gender and age, ethnicity and educational level can be considered important factors [6]. TTH can be defined according to three models: (i) In the kinesiopathological model, the onset and evolution of TTH are due to movement patterns or postural habits. In this model, the term “strain” is associated with biomechanical changes and changes in the contractile capacities of the cervical and cranial muscles, namely, the frontal, temporal, masseter, pterygoid, sternocleidomastoid, splenius, levator scapulae, and trapezius muscles [7,8]. There is now evidence that the origin of this condition is much more complex and cannot be explained by muscle “tension” alone [1,9,10]. (ii) The anatomopathological model involves central and peripheral nervous system mechanisms. At the peripheral level, myofascial trigger points (PGMs) are present in the muscles involved, and there is pain associated with the reproduction of the headache pain pattern in response to stimulation of these PGMs. There is a greater sensitivity to pain in the cervical, cranial and extracranial musculature, which results in a decreased pain detection and tolerance threshold [1,6,11]. (iii) In the psychosocial model, stress is understood as emotional tension. In fact, lifestyles with excessive stress, anxiety, negative feelings and even depression appear as triggering factors of TTHs. Similarly, there seems to be a bidirectional relationship between sleep disorders and TTH, which are considered risk factors for the onset or chronicity of this type of headache [11,12]. In addition to these factors, dehydration, caffeine, and tobacco or alcohol abuse, as well as hormonal fluctuations and bruxism, are other risk factors for the development of TTHs [10,13]. Thus, it is verified that this condition has a great impact on the quality of life of individuals. Irregular and unpredictable pain episodes cause functional limitations and a consequent decrease in productivity, with students showing a 24.4% decrease in productivity during episodes of TTH [14].

The Portuguese Headache Society divides TTH intervention into two types: acute-phase intervention and prophylactic intervention. Acute-phase intervention is recommended during pain episodes (or “crises”) and is based on pharmacological measures, aiming to reduce the intensity of symptoms. Prophylactic intervention aims to prevent the crises or decrease their frequency and intensity through pharmacological and nonpharmacological measures [3,12,15]. In this context, physical therapy acts to reduce the duration, intensity and frequency of crises through the application of manual therapy techniques, neck mobilization, muscle stretching and therapeutic exercise [10,16,17].

In recent years, clinical Pilates has emerged as an option for intervention in musculoskeletal disorders, increasingly used by physical therapists [18,19]. This method is based on 8 principles and consists of 34 original exercises (“mat work”) that promote central stabilization, flexibility, endurance, and postural and body awareness and correct muscular imbalances [20]. Clinical Pilates builds on these 34 exercises by adapting them for specific individual intervention or through classes for individuals with similar conditions [21,22]. Thus, the aim is to verify the impact of clinical Pilates exercises on pain intensity, impact headaches, neck disability, state of anxiety, depression, and stress and on the quality of sleep, in a case series, in higher-education students.

## 2. Materials and Methods

### 2.1. Study Design and Ethics

An observational study of the case series type was conducted. The population of this study consisted of individuals with TTH. Based on this population and to meet the objectives of the study, a convenience sample was selected, composed of students from the Escola Superior de Saúde do Vale do Sousa (ESSVS) and the Instituto Universitário de Ciências da Saúde (IUCS). The study was approved by the Ethics Committee under registration no. 29/CE-IUCS/2020. All participants signed the informed consent, in accordance with the Declaration of Helsinki, and were informed about the objectives and procedures and the guarantee of data confidentiality.

### 2.2. Sample Recruitment and Eligibility Criteria

Initially, a questionnaire was disseminated by email to 2444 ESSVS and IUCS students, to which 149 responses were obtained. Of these, 48 individuals were eligible to participate in the study according to the ICHD-3 TTH diagnosis criteria, and subsequently only 10 individuals showed availability to participate in the study. During the 2 weeks of intervention, one of the participants attended only the first session, reducing the sample to 9 individuals (Figure 1).

The inclusion criteria were defined as: having TTH according to the ICHD-3 criteria and being available to participate in 4 sessions of clinical Pilates. The exclusion criteria were: having had a therapeutic approach for TTH intervention less than 1 month ago and having any type of therapeutic approach for TTH intervention during the study period [23].

### 2.3. Measures

The final questionnaire was made available online and consisted of different sections. The first section aimed to collect sociodemographic data. The second and third sections made it possible to determine the eligibility of individuals through the characterization of headaches, according to the ICHD-3, and these were accessible only to individuals who reported having headaches. At the end of the questionnaire, information was requested about the availability to perform the clinical Pilates sessions in person at the ESSVS facilities. The Numerical Rating Scale (NRS), the Headache Impact Test (HIT-6), the Depression, Anxiety, Stress Scales (DASS-21), and the Pittsburgh Sleep Quality Index (PSQI-PT) were also used. All of them were validated for the Portuguese population.

#### 2.3.1. Sociodemographic Questionnaire

The questionnaire was designed by the main investigators with the aim of characterizing the sample and collecting sociodemographic data.

A pilot study was conducted in order to analyze the sample selection questionnaire and identify possible problems in its content and understanding by individuals. To this end, the questionnaire was made available online in Google Forms format for 4 days to 9 individuals with characteristics similar to those of the sample under study.

#### 2.3.2. Data Collection

Prior to the beginning of the intervention, between 26 March and 24 April 2022, the online sample selection questionnaire was made available to verify the eligibility of individuals to participate in the study, according to the inclusion and exclusion criteria. After 5 days, individuals eligible for the study were contacted by email to confirm their availability to participate in the clinical Pilates sessions. Of the 20 individuals who had responded that they were available to participate in the study, only 10 confirmed their availability in response to the email, and these responses allowed the scheduling of the clinical Pilates sessions. Thus, sessions were organized with different schedules, allowing the participation of all available individuals.

#### 2.3.3. Intervention

Before the beginning of the first session, the participants filled in the questionnaires for measuring the study outcomes (NRS, HIT-6, NDI, and PSQI-PT) and were asked to answer based on the last 2 weeks, except for the DASS-21 where the answers were based only on the last week. The classes aimed to: (i) improve the movement pattern, particularly of the cervical spine and shoulder girdle; (ii) raise awareness of a good posture to prevent further TTH episodes; and (iii) promote muscular and psychological relaxation.

The clinical Pilates classes were taught by the main investigators because they are trained in clinical Pilates. These classes took place at the ESSVS facilities during May 2022 and were divided into 4 sessions of 30 min with an interval of 2 to 4 days between each session. The clinical Pilates sessions are described in Table 1. It should be noted that the first session integrated mainly the learning of the 5 key elements of clinical Pilates that are essential to good practice. The selection of exercises for the other sessions took into consideration the involvement of the local muscles of the different regions, with a holistic intervention perspective.

The outcome evaluation questionnaires were applied again at the end of the fourth session.

#### 2.3.4. Data Processing

The Microsoft Excel program was used to present the results. Thus, data on sample characterization and all instruments used were entered into a Microsoft Excel spreadsheet, and the relative frequencies, the percentage of variation and descriptive measures such as mean and standard deviation were calculated, which allowed characterizing the sample and the variables under study.

## 3. Results

### 3.1. Sample Characteristics

Of the 149 individuals who responded to the questionnaire, 48 (32.2%) reported episodes of TTH: 22 (45.8%) had infrequent episodic TTH, 24 (50%) frequent episodic TTH, and 2 (4.2%) chronic TTH. It was observed that 33 (68.8%) were women and that ages ranged from 18 to 49 years, corresponding to a mean (sd) age of 22.7 (4.98) years.

Of the 10 individuals who participated in the study, 8 (80%) were female, and their ages ranged from 21 to 27 years, corresponding to a mean (sd) age of 23.2 (2.35) years. Regarding the course of study, 5 (50%) individuals are studying for a degree in physical therapy, 4 (40%) are studying for an MSc in dental medicine, and 1 (10%) is studying for an MSc in veterinary medicine.

Table 2 shows the results of the sample characterization regarding the diagnostic criteria (A, B, C and D) for TTHs from the ICHD-3.

Taking Table 2 into consideration, the specific characteristics of TTH were analyzed based on the ICHD-3 diagnostic criteria, and a higher proportion of individuals (40%) with frequent episodic TTH was observed. In addition to the latter type, 30% of individuals with probable frequent episodic TTH, 20% with infrequent episodic TTH, and 10% with probable infrequent episodic TTH were found.

### 3.2. Outcomes

Table 3 shows the percentages of variation of the scores obtained in the 5 study outcomes between the initial and final evaluation for everyone.

#### 3.2.1. Numerical Rating Scale (NRS)

By analyzing the results obtained, we can see that the majority (77.78%) of the participants (B, C, D, E, F, H, and I) did not present any change between the initial and final evaluation regarding the intensity of headache pain, with only 2 participants (individuals A and G) showing an improvement with a variation of −10%.

#### 3.2.2. Headache Impact Test (HIT-6)

Through the percentage of variation, we see an improvement in quality of life in relation to headache for all participants at the end of the intervention, ranging from −1.28% to −23.08%.

#### 3.2.3. Neck Disability Index (NDI)

From the results obtained, 8 of the 9 participants improved, with a score variation between −2% and −18%, and only one participant (D) had no improvement.

#### 3.2.4. Depression, Anxiety, Stress Scales (DASS-21)

By analyzing the results obtained, we can see that 6 participants (B, C, E, F, H, and I) showed a positive evolution. The participants with positive evolution had percentages of variation between the initial and final moments from −1.59% to −9.52%.

#### 3.2.5. Pittsburgh Sleep Quality Index—Portuguese Version (PSQI-PT)

From the results obtained during the initial assessment, 8 participants (A, B, C, E, F, G, H and I) improved. Of those who improved, the percentages of variation presented varied between −4.76% and −14.29%, with the individuals with the greatest improvement being E, F, and I. Finally, individual D got worse, with a variation percentage of +9.52%.

Figure 2 shows the results of the sum of the scores of each DASS subscale, allowing comparison of the mean levels of depression, anxiety and stress, as well as their respective evolution. The mean for each participant was considered according to the subscales.

## 4. Discussion

This study aimed to determine the effectiveness of a clinical Pilates exercise program, performed in class, in the intervention of TTH, in students of the Cooperative of Polytechnic and University Higher Education (CESPU).

All classes were performed under supervision, with individual correction whenever necessary, in order to potentiate the effects of clinical Pilates on the outcomes under study.

The outcomes assessed, relative to the THT, were pain intensity and the impact of the THT, functional disability due to pain in the cervical region, psychological state (in regard to stress, anxiety, and depression), and quality of sleep, and it was found that most individuals had a positive evolution except for pain intensity.

The prevalence of TTH in the sample of the present study was 32.2%. A study also conducted in students by Kaynak Key et al. (2004) [24] showed a lower prevalence of HTT, but the mean age was lower than that in the present study. However, Mannix (2001) [25] and Steele et al. (2021) [6] reported an increasing prevalence of HTT as a function of age up to a peak prevalence between 30 and 39 years of age, which corroborates the results of the present study. Furthermore, with a more detailed analysis of the individuals in the sample, a higher prevalence of episodic THT than of chronic THT was observed. These results are slightly higher than those mentioned by Mannix (2001) [25], who demonstrated a prevalence of chronic HTT of 2.2% in the general population. This difference in prevalence may be explained by changes in emotional state, which may predispose to episodic TTH.

Regarding gender, there was a higher percentage of women with HTT (68.8%) than men (31.3%). These results correspond to those described by Torelli et al. (2004) [23], with percentages of 69% for women and 31% for men, in a sample of 48 individuals with THT. One of the reasons that could explain these differences would be hormonal fluctuations, with women being more susceptible to these variations.

Reducing the intensity and frequency of pain attributed to the headache (as well as the duration of the crises) appears to be one of the greatest challenges in the therapeutic approach to THT. In the present study, it was observed that most participants did not show improvement regarding the intensity of pain perceived during headache episodes. The literature regarding whether or not clinical Pilates is effective in this outcome is scarce. However, these results can be compared with those of the study conducted by Torelli et al. (2004) [23], which aimed to evaluate the effect of physical therapy (namely, a therapeutic exercise program directed at the muscles of the cervical and pericranial regions) as a prophylactic intervention for TTH. The conclusion was that physiotherapy and therapeutic exercises had no influence on pain intensity and seizure duration but did influence seizure frequency. However, this information is currently still controversial, and studies have also been found in which therapeutic exercises seem to influence a decrease in pain intensity in individuals with THT [26,27]. It should also be noted that therapeutic exercises have more significant long-term benefits [28].

As previously mentioned, THTs influence the quality of life of individuals in several activities of daily living. When we compared the scores obtained through the HIT-6 at the two moments of assessment in this study, we found a decrease in the impact of headache on quality of life in all participants. These results should be treated with caution, considering that no studies were found in the literature to confirm or contradict this trend of improvement in the impact of headache on the quality of life of individuals with THTs in relation to clinical Pilates. However, the studies by Hosseinifar et al. (2016) [27] and Sertel et al. (2017) [26] also report a significant decrease in the impact of headache on quality of life following the implementation of a therapeutic exercise program.

Considering the involvement of muscles of the cervical region in the pathogenesis of THT and the frequent association of this condition with pain in the cervical region, it seems important to assess the functionality of this area [29]. According to the results obtained in this study, most participants improved regarding the functional disability due to pain in the cervical region, which is corroborated by the study of Mallin and Murphy (2013) [30], in which a significant improvement was observed in the results of this parameter. Several studies have also shown that clinical Pilates, due to its holistic effect, presents positive and significant results regarding the reduction of stress, anxiety, and depression in healthy individuals or those with clinical conditions [31,32,33,34,35]. The results observed in the present study show us that with only 4 sessions, 66.7% of the individuals showed an overall improvement in their psychological state, with an increase only in the values related to anxiety. Thus, the mean values obtained regarding the decrease in stress and depression after the intervention corroborate the results of the studies cited above. One hypothesis to explain this result may have been the influence of the academic period (assessment period) in which students were at the time of the intervention. In fact, studies conducted by Khoshhal et al. (2017) [36] report high levels of anxiety in this period in the student population, compared with stress and depression. 

In addition to psychological symptoms, sleep regulation is a complex mechanism in which several factors such as hormonal fluctuations or mood disturbances are involved [37]. According to Caldwell et al. (2009) [38], half of the student population also has sleep disorders. This information reinforces, in addition to the bidirectional relationship between TTH and sleep [12], the importance of studying the evolution of sleep quality in relation to intervention in a population of students with TTH. From the results obtained in the present study, an improvement in sleep quality is observed in most participants, corroborating other studies that report similar results in students and workers up to 40 years of age [37,38,39]. Even a study by Lins Filho et al. (2019) [40] suggests that the best stress management and improvement in overall quality of life promoted by clinical Pilates are related to improved sleep quality. Thus, clinical Pilates appears to be a good option for a therapeutic approach that focuses on increasing the strength and stability of the deep muscles, particularly in the cervical region. This ensures that the superficial muscles are not used as compensation [41] and contributes to a better muscle balance through the co-activation of antagonist muscles during movement, which reflects a reorganization of the motor control strategy as a method of protection and adaptation to pain [42]. In addition, when one observes the results of this study and the comparisons made with other studies, the clinical Pilates exercises seem to suggest a favorable evolution in the impact of headache on the quality of life of individuals with THT. In fact, this method brings benefits by increasing body functionality, general health, psychological status and sleep quality [37,39,43].

This study has some limitations, namely, the academic period in which the intervention was carried out, since it may have influenced the results obtained, considering the increased levels of stress, anxiety, and depression in students during assessment periods. The limited literature available on the effects of clinical Pilates on THT implies that our results were analyzed with consideration of the effects of clinical exercise.

As future suggestions, it seems interesting to conduct more scientific studies that address this issue, evaluating other parameters such as muscle activity through electromyography (EMG) analysis, the painful sensitivity of the craniocervical muscles or the measurement of the cranial vertebral angle. The measurement of the cranial vertebral angle seems pertinent, being associated with posture and increased pain in the cervical region.

## 5. Conclusions

After the intervention, results showed that pain intensity decreased only in 2 participants; all improved regarding the impact of tension-type headaches on normal daily life and ability to function; and only one did not show improvement in functional disability caused by pain in the cervical region. In relation to the negative emotional states, 6 individuals reported improvements, and only one individual reported a lower quality of sleep after the program.

## Figures and Tables

**Figure 1 behavsci-13-00105-f001:**
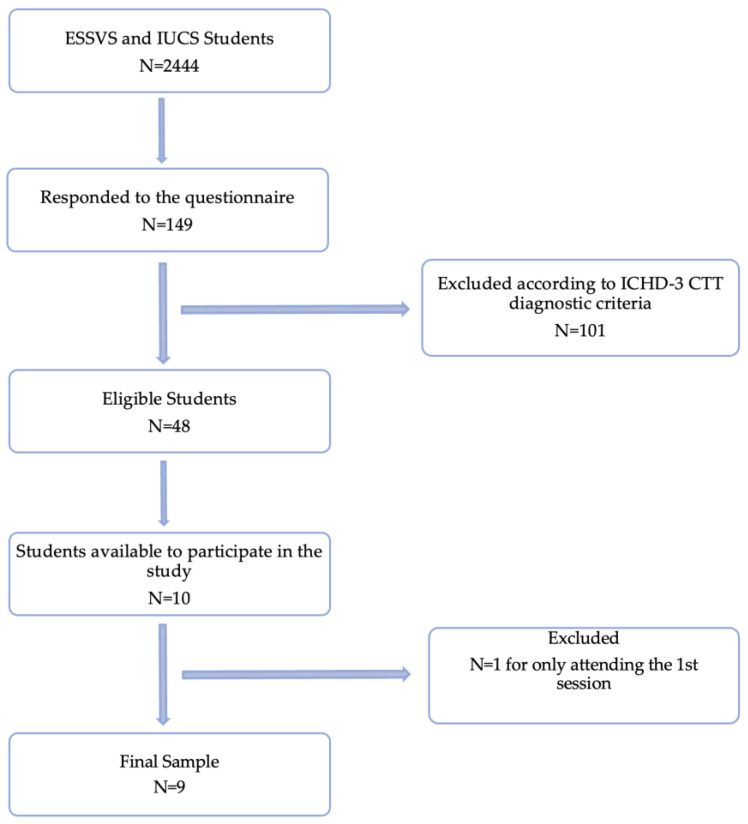
Flowchart of sample selection.

**Figure 2 behavsci-13-00105-f002:**
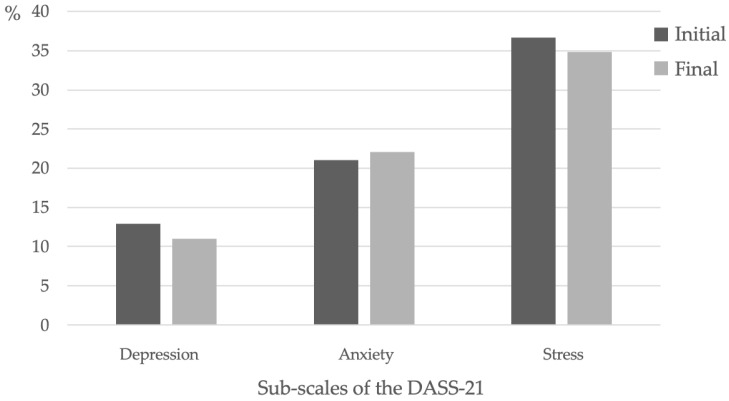
Mean values of the depression, anxiety and stress subscales at the two assessment times.

**Table 1 behavsci-13-00105-t001:** Condition and exercises related to each session of the intervention.

Session Number	Condition of the Session	Exercises
Session 1	Learning the 5 key elements	- Hundred level 1
Session 2	Closed kinetic chain exercises	- Hundred level 1 - Hundred level 2 - Overhead Reach level 1 with variation - Swan Dive level 1 - Breath Stroke Preparation level 1 with variation
Session 3	Open kinetic chain exercises	- Hundred level 1 - Hundred level 3 - Overhead Reach level 1 - Swan Dive level 3 - Breath Stroke Preparation level 2
Session 4	Mobility Exercises	- Hundred level 3 - Arm Openings level 1 - Arm Openings level 2 - Spine Twist level 1

**Table 2 behavsci-13-00105-t002:** Characterization of the sample according to the diagnostic criteria for TTHs of the ICHD-3.

	Criterion A (Presence of the Headache)	Criterion B (Presence of the Causative Disorder)	Criterion C * (Evidence of Causation)	Criterion D (Other Symptoms during Headache)	TTH Classification
Individual A	At least 10 headache episodes occurring on 1 to 14 days on average for more than 3 months	The headache lasts from 30 min to 72 h	It has 3 of the 4 characteristics of criterion C	No associated	2.2. Frequent episodic TTH
Individual B	At least 10 headache episodes occurring on 1 to 14 days on average for more than 3 months	The headache lasts from 30 min to 72 h	It has 4 of the 4 characteristics of criterion C	Photophobia and phonophobia **	2.4.2 Probable frequent episodic TTH
Individual C	At least 10 headache episodes occurring on 1 to 14 days on average for more than 3 months	The headache lasts less than 30 min **	It has 4 of the 4 characteristics of criterion C	No associated	2.4.2 Probable frequent episodic TTH
Individual D	At least 10 episodes of headaches occurring on <1 day per month on average	The headache lasts less than 30 min **	It has 3 of the 4 characteristics of criterion C	No associated	2.4.1 Probable infrequent episodic TTH
Individual E	At least 10 episodes of headaches occurring on <1 day per month on average	The headache lasts from 30 min to 72 h	It has 3 of the 4 characteristics of criterion C	No associated	2.1 Infrequent episodic TTH
Individual F	At least 10 headache episodes occurring on 1 to 14 days on average for more than 3 months	The headache lasts less than 30 min **	It has 4 of the 4 characteristics of criterion C	No associated	2.4.2 Probable frequent episodic TTH
Individual G	At least 10 headache episodes occurring on 1 to 14 days on average for more than 3 months	The headache lasts from 30 min to 72 h	It has 4 of the 4 characteristics of criterion C	Photophobia	2.2. Frequent episodic TTH
Individual H	At least 10 headache episodes occurring on 1 to 14 days on average for more than 3 months	The headache lasts from 30 min to 72 h	It has 3 of the 4 characteristics of criterion C	No associated	2.2. Frequent episodic TTH
Individual I	At least 10 episodes of headaches occurring on <1 day per month on average	The headache lasts from 30 min to 72 h	It has 3 of the 4 characteristics of criterion C	No associated	2.1 Infrequent episodic TTH
Individual J	At least 10 headache episodes occurring on 1 to 14 days on average for more than 3 months	The headache lasts from 30 min to 72 h	It has 3 of the 4 characteristics of criterion C	No associated	2.2. Frequent episodic TTH

* Criterion C refers to ICHD-3 criterion. ** These criteria do not match the diagnostic criteria for TTH, and individuals with these then have a diagnosis of probable TTH.

**Table 3 behavsci-13-00105-t003:** Percentages of variation for the different outcomes.

	% Variation in NRS	% Variation in HIT-6	% Variation in NDI	% Variation in DASS-21	% Variation in PSQI-PT
Individual A	−10	−2.56	−6	+6,35	−4.76
Individual B	0	−6.41	−8	−4.76	−9.52
Individual C	0	−16.67	−6	−6.35	−9.52
Individual D	0	−7.69	0	+4.76	+9.52
Individual E	0	−23.08	−12	−7.94	−14.29
Individual F	0	−2.56	−18	−3.17	−14.29
Individual G	−10	−1.28	−2	+15.87	−4.76
Individual H	0	−1.28	−12	−9.52	−61.90
Individual I	0	−3.84	−6	−1.59	−14.29

## Data Availability

The data presented in this study are available on request from the corresponding author. The data are not publicly available due to the accordance to the confidentiality agreement with the company does not allow the public disclosure of the data available.

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
