# Peer review of "The Impact of Clinical Pilates Exercises on Tension-Type Headaches: A Case Series"

_behavsci, 2023, doi:10.3390/bs13020105_

Round 1
Reviewer 1 Report
Reviewer's report
Title: The impact of Clinical Pilates exercises on Tension-type Head-aches: a case series
Reviewer's report:
Leite and colleagues examined the impact of Clinical Pilates exercises on Tension-type Head-aches through a case series.
This is an interesting paper that explores a few studied theme; however, a some issues need to be addressed.
Title
Nothing to report.
Abstract
The sentence is not clear. (Page ; line 14 and 15).
Should be interesting to refer it was a group class intervention.
Keywords: shouldn’t “Pilates” be on the keywords
Introduction
Well written and literature based.
The aim of this study should refer that it is made in a case series methodology.
Materials and Methods
Were all instruments validated for the Portuguese population?
How was the Pilates exercise selection made?
Results
Is table 2 really needed? This information seem to be repeated in the text below.
I have some problems in understanding table 3. Check also the English language.
Do you think it could be useful to gather all the information in a single table that can make it easier for readers to understand all the characteristics and changes produced by the intervention
What do you mean by “average percentage” in figure 1?
Discussion
Please the differences between class group versus individual sessions; The exercise selection was important in the effect?
Are 4 sessions enough to produce short-term changes in the participants?
Is the “impact” produced by Exercise or Pilates? This sometimes is not clear during the discussion
Conclusion
The last sentence is not clear. Please try to be more concise in the conclusions obtained with the study.
Level of interest: An article of interest
Quality of written English: Good
Declaration of competing interests: no competing interests
Author Response
We would like to thank the reviewer for reading our paper carefully. The valuable comments and questions have allowed us to significantly improve our manuscript.
Reviewer 2 Report
Overall, the article is quite well written. However, referring to such a common problem, I would expect a much larger number of respondents, more statistical data. I lack a description of the possible mechanisms of the influence of exercises on the described and studied problem. I believe that the article needs more attention and more people surveyed.
Author Response

(The authors gave the same response as above.)
